Bidirectional association between breast cancer and dementia: a systematic review and meta-analysis of observational studies

Bao Fuxing
Yu Liang yuliang_1208@qq.com
Zhang Xiaolei
Mu Qier
Department of Ultrasound, Inner Mongolia Maternity and Child Health Care Hospital , Hohhot, Inner Mongolia , China
Tyagi Abhishek
Electronic publication date: 2025 Jan 31
Publication date: 2025
Volume: 13
Electronic Location ID: e18888
Received 2024 Sep 13; Accepted 2024 Dec 31
Copyright: © 2025 Bao et al.
Copyright year: 2025
Copyright holder: Bao et al.
License: This is an open access article distributed under the terms of the Creative Commons Attribution License, which permits unrestricted use, distribution, reproduction and adaptation in any medium and for any purpose provided that it is properly attributed. For attribution, the original author(s), title, publication source (PeerJ) and either DOI or URL of the article must be cited.
License URL: https://creativecommons.org/licenses/by/4.0/

Keywords: Breast cancer survivors, Dementia, Observational study, Meta-analysis

Funding: There was no funding for this work.

==============================
Background

Cognitive decline following cancer treatment can vary from mild cognitive impairment to severe dementia. However, there is inconsistent evidence regarding the relationship between breast cancer survivors and their risk of developing dementia. This meta-analysis aims to consolidate observational studies to explore the bidirectional association between breast cancer and dementia risk.

Methods

We conducted a comprehensive search using medical subject headings (MeSH) and keywords across PubMed, Cochrane Library, and Embase databases to identify cohort, case-control, and cross-sectional studies examining the link between breast cancer and dementia risk. Statistical analysis was performed using Stata version 14.0, with a random effects model employed to account for heterogeneity. Publication bias was assessed using funnel plots and Egger’s test.

Results

This meta-analysis included 13 studies with a total of 346,051 participants, up to June 20, 2024. Of these, seven studies investigated the risk of dementia among patients with breast cancer, revealing a lower risk [OR = 0.56, 95% CI [0.27–1.18], I2 = 99.1%, P = 0.128]. Similarly, seven studies explored the risk of breast cancer in individuals with dementia, showing a lower risk as well [OR = 0.79, 95% CI [0.51–1.22], I2 = 94.5%, P = 0.290].

Conclusion

Our findings indicate that breast cancer is less likely to lead to dementia and that dementia is similarly low associated with risk of breast cancer. These insights are crucial for clinicians in guiding the prevention and monitoring of neurodegenerative conditions in patients with breast cancer.

Introduction

In 2020, breast cancer emerged as the most prevalent malignancy globally, accounting for the highest incidence and mortality rates among women. Specifically, breast cancer constitutes 12.50% of all cancer cases and 6.92% of cancer-related deaths (2021–2023). Data from the Global Burden of Disease (GBD) indicated a significant increase in breast cancer incidence in the Western Pacific region (WPR) over the past 30 years, with the number of cases rising from 143,889.3 to 551,442.2. This upward trend is projected to continue from 2020 to 2044. Age is a critical factor in breast cancer development, with the risk notably increasing after the age of 40 (Wu et al., 2024). Fortunately, advancements in early detection and tailored therapeutic interventions have led to substantial improvements in survival rates. Currently, the 5- to 10-year survival rate for breast cancer patients ranges from 85% to 90% (Nardin et al., 2020). Despite these positive developments, many women who are diagnosed and treated for breast cancer experience significant short-term and long-term consequences, including cancer-related cognitive impairment (CRCI) (Carreira et al., 2021; Chapman, Derakshan & Grunfeld, 2023). Studies have documented that patients with breast cancer undergoing chemotherapy may experience cognitive disorders that persist for up to two decades (Koppelmans et al., 2012).

Cognitive decline following cancer treatment can vary from mild cognitive impairment to severe dementia. Dementia is characterized by acquired cognitive impairments that significantly affect a patient’s thinking and daily functioning (GBD 2019 Dementia Forecasting Collaborators, 2022). This condition typically develops gradually, initially impacting daily activities and eventually limiting social interactions. With improved treatment outcomes and increased lifespan among patients with breast cancer, the incidence of cognitive impairment, Alzheimer’s disease, dementia, and other neurodegenerative disorders has also risen significantly (Du, Xia & Hardy, 2010). Identifying risk factors for cognitive impairment is thus crucial for the early detection of high-risk cancer and for developing strategies to alleviate the medical burden in an aging population.

Kao, Yeh & Chen (2023) reviewed literature on cognitive impairment across various cancer types, including lung, breast, head and neck, gastric, prostate, colorectal, and brain tumors/metastases, assessing the impact of different treatments. Breast cancer treatments include surgery, chemotherapy, hormone therapy, and radiotherapy, with chemotherapy being a core component aimed at targeting proliferating cells. However, chemotherapy can adversely affect the central nervous system, leading to cognitive impairment in a significant number of patients, both during and after treatment. Long-term studies indicate that 17% to 75% of breast cancer may experience impairments in attention, concentration, planning, and working memory even 6 months to 20 years post-chemotherapy (Ahles, Root & Ryan, 2012).

There is ongoing debate regarding the association between cancer and dementia risk. Some community-based cohort studies suggest that cancer diagnosis may be linked to a lower pathological burden of Alzheimer’s disease and reduced cognitive impairment (Karanth et al., 2022), a finding also reported in breast cancer (Oh et al., 2023). Conversely, other studies indicate that breast cancer is associated with an increased risk of dementia (Baxter et al., 2009; Blanchette et al., 2020; Bromley et al., 2019; Carreira et al., 2021; Raji et al., 2009; Thompson et al., 2021; Wennberg et al., 2023). Thus, a systematic review and meta-analysis are warranted to elucidate the relationship between breast cancer and dementia risk, providing valuable insights for identifying risk factors and developing preventive strategies.

Methods

Study designs

This systematic review was conducted following the guidelines outlined by the Preferred Reporting Items for Systematic Reviews and Meta-Analyses (PRISMA) (Page et al., 2021) and the Meta-analysis of Observational Studies in Epidemiology (MOOSE) guidelines (Stroup et al., 2000). Additionally, this review has been registered with the International Prospective Register of Systematic Reviews (PROSPERO) under the approval number CRD42023443702.

Data sources

We performed a comprehensive search of PubMed, EMBASE, and the Cochrane Library from inception up to June 20, 2024, without any restrictions. The search utilized subject terms (Emtree in EMBASE, MeSH in PubMed) and relevant keywords related to dementia, cognitive impairment, Alzheimer’s disease, and breast cancer, including the variants. We also reviewed reference lists of retrieved studies and previous meta-analyses (Lei et al., 2022; Li et al., 2023; Wang et al., 2022; Wang, Sang & Zheng, 2022; Wang, Sang & Zheng, 2024; Zhang et al., 2023) to identify additional studies that might be eligible for inclusion. The full search strategy was detailed in Tables S1–S3.

Eligibility criteria

Inclusion criteria:

(a) Population: Individuals diagnosed with either dementia or breast cancer, with no restrictions on age, race, or gender.

(b) Exposures: Patients who have breast cancer, regardless of disease stage, treatment, or pathological type, or patients with various forms of dementia (e.g., vascular dementia, senile dementia, Alzheimer’s disease, and mixed dementia). Studies must investigate the association between dementia and the risk of breast cancer or vice versa.

(c) Comparison: Individuals without breast cancer or dementia.

(d) Outcomes: Newly developed cases of dementia or breast cancer, with reported estimates including hazard ratios (HRs), relative risks (RRs), or odds ratios (ORs) and 95% confidence intervals (95% CIs).

(e) Study Types: Prospective or retrospective cohort studies, or case-control studies.

Exclusion criteria:

(a) Conference abstracts or letters to editors.

(b) Studies with overlapping data or duplicate publications.

(c) Incomplete reporting or studies not addressing the outcomes of interest.

Study selection

Retrieved records were imported into Endnote reference management software. Duplicate records were identified and removed using both the software’s duplicate checking function and manual review. Two reviewers (FX Bao and L Yu) then evaluated titles and abstracts to exclude studies that did not meet the predefined inclusion criteria. Full-text articles were reviewed, with detailed justifications for exclusion provided according to AMSTAR 2 specifications (Shea et al., 2017). Exclusion reasons were documented in the PRISMA literature screening flowchart. A third reviewer (QE Mu) verified the findings and resolved any discrepancies through discussion.

Data extraction

Two reviewers (FX Bao and L Yu) independently extracted data using pre-designed Excel forms based on meta-analysis data extraction guidelines (Taylor, Mahtani & Aronson, 2021). Extracted information included author, publication year, country, study type, sample size, study period, follow-up duration, participant age, diagnoses of breast cancer and dementia, and adjusted confounders. Any discrepancies were resolved through discussion with QE Mu.

Quality assessment

The quality of cohort studies was assessed using the Newcastle-Ottawa Scale (Available from: http://www.ohri.ca/programs/clinical_epidemiology/oxford.asp). This scale includes nine stars for participant selection and exposure measurement, two stars for result comparison, and three stars for outcome assessment and follow-up adequacy. Higher scores indicate better quality, with scores of 0–3 considered low quality, 4–6 moderate quality, and 7–9 high quality.

Data synthesis

Given the restricted outcome measures across all populations and subgroups, distinctions among OR, HR, RR, and IRR were generally considered negligible (Wu et al., 2021). Study-specific OR/HR/RR/IRR were aggregated and transformed into OR with corresponding 95% CI for meta-analysis. Adjusted ORs and 95% CIs were used to evaluate the relationship between dementia incidence and breast cancer. Heterogeneity was assessed using I2 values. To address heterogeneity and confounding factors, a random effects model was employed (Chen et al., 2020; Ning et al., 2023). Sensitivity analysis was performed by pre-defining criteria for study exclusion to ensure robustness of the overall effect (Higgins et al., 2003). Funnel plots were used to detect publication bias, and Egger regression tests provided statistical evaluation (Egger et al., 1997). Pruning and filling methods were applied as needed to assess potential biases. All statistical analyses were conducted using Stata Corp, College Station, Texas statistical software version 14.0.

Subgroup analyses for breast cancer were considered based on different pathological types, diagnostic criteria, survival times, and other factors, provided sufficient data were available (Chapman, Derakshan & Grunfeld, 2023; Peifer & Gass, 2022; Tucholka et al., 2018). Similarly, subgroups for dementia risk included various types of dementia, cognitive impairment, and Alzheimer’s disease, contingent upon the availability of sufficient original data.

Confidence in evidence

The Grading of Recommendations Assessment, Development and Evaluation (GRADE) system (Guyatt et al., 2011) was used to assess the quality of evidence in observational studies. Initially, evidence from observational studies is rated as low quality, but can be upgraded based on significant effect sizes (OR ≥ 2 or ≤ 0.5), dose-response gradients, or the exclusion of factors that may degrade quality (Guyatt et al., 2011a, 2011b, 2011c). Evidence is categorized into four levels: high, medium, low, or very low.

Ethics and dissemination

This meta-analysis utilized data from previously reported studies, and did not involve direct participation of patients or populations. Therefore, ethical approval was not required. The results of this meta-analysis will be published in peer-reviewed journals to facilitate dissemination and review.

Results

Literature search

A total of 4,778 records were collected through the search. After title and abstract screening, 84 articles were considered potentially relevant. Thirteen studies were included after full text review, which reported the risk of dementia or breast cancer. The selection process was presented in Fig. 1.

Figure 1 PRISMA flow diagram.

Study characteristics

This meta-analysis encompassed 13 studies, enrolling a total of 346,051 individuals, with a publication date up to June 20, 2024 (Attner et al., 2010; Baxter et al., 2009; Heun et al., 2013; Jørgensen et al., 2012; Khan et al., 2011; Kurita et al., 2017; Musicco et al., 2013; Oh et al., 2023; Ording et al., 2019; Ou et al., 2013; Ren et al., 2022; Roderburg et al., 2021; Valentine et al., 2022). Of these studies, ten were cohort investigations, two were case-control studies, and one was prospective study type. The primary characteristics of the studies included in this analysis were detailed in Table 1, and the excluded studies were shown in Table S4.

Table 1 Characteristics of the included studies.

Author	Year	Country	Study period	Case group	Female/Male	Study type	Follow-up years	Diagnostic criteria	Control type	Age (Mean ± SD)	Effect value type	Outcome	
Oh J	2023	Korean	January 2009 to December 2018	2,252	NA	Cohort study	8.67 ±2.0	ICD-10	Without breast cancer	66.27 ± 7.32; 66.28 ± 7.37	HR	1	
Roderburg C	2021	Germany	January 2000 and December 2018	11,708	11,708/0	Cohort study	6.8 ± 6.2	ICD-10	Non cancer	74.8 ± 6.4	HR	1	
Kurita GP	2017	30 centers in 12 countries	April 2011 to October 2013	1,568	780/788	Prospective study	4 to 16 weeks	NA	NA	65.5	OR	1	
Musicco M	2013	Milano	January 1, 2004 to December 31, 2009	21,451; 2,832	12,225/9,226; 947/1,885	Cohort study	NA	ICD-10	The general population	72.4 ± 7.8; 78.1 ± 6.8	RR	1; 2	
Jørgensen TL	2012	Denmark	1 January 1996 to 31 December 2006	6,325	NA	Cohort study	NA	ICD-10	Controls	NA	OR	1	
Khan NF	2011	The UK	1 September 2003 to 31 August 2006	84,587	84,587/0	Cohort study	More than 5 years	NA	Controls	66.9 ± 12.3	HR	1	
Baxter NN	2009	The US	January 1, 1992 to December 31, 1999	21,362	21,362/0	Cohort study	4.92	ICD-9	NA	73	HR	1	
Ren RJ	2022	China	July 1, 2013 to January 31, 2016	8,097	4,753/3,344	Cohort study	0.01–6.46	ICD-10	Without AD	83 (79, 86)	HR	2	
Valentine D	2022	The US	1994 to 2014	85,787	41,054/44,733	Cohort study	NA	ICD 9 and ICD 10	UPDB population	NA	RR	2	
Ording AG	2019	Danish	1980 to 2012	72,732	46,186/26,546	Cohort study	2.8 (1.1, 5.0)	NA	NA	81 (74, 107)	SIR	2	
Heun R	2013	The UK	1 January 2000 to 31 December 2007	634	413/221	Case-control study	7	ICD-10	Control subjects	85.1 (8.2); 80.8 (7.4)	RR	2	
Ou SM	2013	China	March 1, 1995 to December 31, 2009	6,960	4,198/2,762	Cohort study	4.25 (2.81–5.92)	NA	The national	76 (70–80)	SIR	2	
Attner B	2010	Sweden	2005 to 2007	19,756	NA	Case-control study	NA	ICD 10	NA	NA	RR	2	
Note:

12 countries: Australia, Bulgaria, Belgium, Canada, Denmark, Georgia, Italy, Norway, Portugal, Spain, Switzerland, and UK; ICD: international classification of diseases; HR, hazard ratio; OR, odds ratio; RR, relative risk; SIRs, standardized incidence rate ratios; Outcome, 1 the risk of breast cancer in dementia, 2 the risk of dementia in breast cancer (Oh et al., 2023; Roderburg et al., 2021; Kurita et al., 2017; Musicco et al., 2013; Jørgensen et al., 2012; Khan et al., 2011; Baxter et al., 2009; Ren et al., 2022; Valentine et al., 2022; Ording et al., 2019; Heun et al., 2013; Ou et al., 2013; Attner et al., 2010).

Quality assessment

According to NOS criteria, the average score was 7.08 of all included cohort and case-control studies, and the score for each trail was 5 or above, indicating that the cohort and case-control studies were of moderate, or high quality in this meta-analysis. The scores of the included cohort and case-control studies were shown in Table 2.

Table 2 Details of the NOS (Oh et al., 2023; Roderburg et al., 2021; Kurita et al., 2017; Musicco et al., 2013; Jørgensen et al., 2012; Khan et al., 2011; Baxter et al., 2009; Ren et al., 2022; Valentine et al., 2022; Ording et al., 2019; Heun et al., 2013; Ou et al., 2013; Attner et al., 2010).

Study	Year	Selection	Comparability	Outcome	Overall quality score	
Oh J	2023	★★★★	★★	★★	8	
Roderburg C	2021	★★★★	★★	★★	8	
Kurita GP	2017	★★		★★	4	
Musicco M	2013	★★★★	★★	★★	8	
Jørgensen TL	2012	★★★★	★★	★★	8	
Khan NF	2011	★★★★	★★	★★	8	
Baxter NN	2009	★★★		★★	5	
Ren RJ	2022	★★★★	★★	★★	8	
Valentine D	2022	★★★★		★★	6	
Ording AG	2019	★★★		★★	5	
Heun R	2013	★★★★	★★	★★	8	
Ou SM	2013	★★★★	★★	★★	8	
Attner B	2010	★★★★	★★	★★	8	

Breast cancer and the risk of dementia

There were seven studies report the risk of breast cancer associated with dementia (Baxter et al., 2009; Jørgensen et al., 2012; Khan et al., 2011; Kurita et al., 2017; Musicco et al., 2013; Oh et al., 2023; Roderburg et al., 2021). The pooling analysis showed that patients with breast cancer reveal a lower risk of dementia (OR = 0.56, 95% CI [0.27–1.18], I2 = 99.1%, P = 0.128, Fig. 2). Sensitivity analysis showed that none of the individual studies reversed the pooled-effect size, which means that the results are robust (Fig. S1, and Table S5).

Figure 2 Meta-analysis of the risk of dementia associated with breast cancer (Oh et al., 2023; Roderburg et al., 2021; Kurita et al., 2017; Musicco et al., 2013; Jørgensen et al., 2012; Khan et al., 2011; Baxter et al., 2009).

Dementia and the risk of breast cancer

There were seven studies mentioned the risk of dementia associated with breast cancer (Attner et al., 2010; Heun et al., 2013; Musicco et al., 2013; Ording et al., 2019; Ou et al., 2013; Ren et al., 2022; Valentine et al., 2022). In the total OR, we were not included one study for not mentioned the overall OR (Ording et al., 2019). The result suggest that individuals with dementia showed a lower risk of breast cancer (OR = 0.79, 95% CI [0.51–1.22], I2 = 94.5%, P = 0.290, Fig. 3). Sensitivity analysis showed that none of the individual studies reversed the pooled-effect size, which means that the results are robust (Fig. S2, and Table S6).

Figure 3 Meta-analysis of the risk of breast cancer associated with dementia (Ren et al., 2022; Valentine et al., 2022; Heun et al., 2013; Musicco et al., 2013; Ou et al., 2013; Attner et al., 2010).

Evidence certainty

The GRADE level of evidence was very low for the risk of dementia with breast cancer. The GRADE level of evidence was also very low for the risk of breast cancer with dementia. GRADE evidence certainty for the outcomes was shown in Table 3.

Table 3 Certainty of evidence and summary effect estimates assessed by GRADE of the study outcomes.

Outcomes	Exposure	Study of findings	Quality assessment	Certainty of
evidence	
No. studies	OR [95% CI]	Study design*	Inconsistency†	Indirectness	Imprecision	Other consideration	
Dementia	Breast cancer	7	0.56 [0.27–1.18]	Serious	Serious	Not serious	Not serious	Not serious	Very low	
Breast	Dementia	7	0.79 [0.51–1.22]	Serious	Serious	Not serious	Not serious	Not serious	Very low	
Note:

Explanations: *Downgraded by one level if >25% of participants in this comparison were from studies at high risk of bias. †Downgraded by one level if heterogeneity (I2)>50%.

Publication bias

A visual inspection of the funnel plot showed no significant publication bias in the risk of dementia and breast cancer (Fig. S3). Egger’s test shown the included studies were completely unbiased (P = 0.272). It also showed no significant publication bias in the risk of breast cancer and dementia and confirmed by Egger’s test (P = 0.239) (Fig. S4).

Discussion

This meta-analysis included 13 studies with a total of 346,051 individuals, providing a thorough evaluation of the bidirectional relationship between breast cancer and dementia. Our findings suggest a decreased risk of dementia among individuals with breast cancer compared to non-breast cancer controls, implying that breast cancer may not be a significant risk factor for dementia. Similarly, the risk of breast cancer appeared to be reduced in individuals with dementia, suggesting that dementia might not be a risk factor for breast cancer either. The population studied, predominantly over 60 years of age, may indicate that the low co-occurrence of these disorders is characteristic of older adults.

Previous research has reported that the presence of both cancer and dementia can lead to diminished cancer-specific and overall survival rates (Kimmick et al., 2014; Möhl et al., 2021), and approximately 7–10% of breast cancer patients also experience cognitive impairment or dementia (Ahmad et al., 2024). The co-existence of dementia impacts cancer treatment and was associated with higher mortality rates for breast cancer patients (Ewertz et al., 2018; Raji et al., 2008). Neurotoxicity from cancer treatments, such as systemic chemotherapy, is common and can affect both central (e.g., headache, seizures, encephalopathy) and peripheral nervous systems (e.g., peripheral neuropathy, plexopathy), which may contribute to dementia. Additionally, adjuvant hormonal therapies for breast cancer, such as corticosteroids and tamoxifen, can impact cognitive function and prognosis (Albers, Kieffer & Schagen, 2024; Brunner et al., 2005; Liao, Lin & Lai, 2017). Cognitive impairments, often referred to as “chemo brain”, might suggest a connection between these two conditions (Chen et al., 2019). In patients with dementia, increased apoptosis may reduce the likelihood of developing breast cancer. Various biological processes, including systemic inflammation, oxidative stress, and disrupted brain structures, may suggest an association between dementia and breast cancer (Nixon, 2017; Nudelman et al., 2019; Shafi, 2016). BRCA1, a key tumor suppressor, has been linked to both dementia and neuronal cell death (Wezyk et al., 2018). Furthermore, mechanisms that lead to DNA damage, such as oxidative stress and deficient DNA repair, are significant in both conditions, potentially indicating a shared pathophysiology. Understanding these mechanisms could be crucial for predicting disease progression and developing new treatments.

Interestingly, our study found a reduced incidence of dementia associated with breast cancer and vice versa. This result may be due to biological factors or diagnostic biases, such as patients being unable to communicate breast cancer symptoms or physicians and families potentially neglecting to investigate symptoms in dementia patients.

The strength of our meta-analysis lied in the inclusion of 13 relevant observational studies, offering a comprehensive view of the relationship between breast cancer and dementia. We provide the first detailed examination of this mutual relationship, exploring both the likelihood of dementia in individuals with breast cancer and the risk of breast cancer in those with dementia. However, our study has limitations. First, the studies included in our research often conflate patients currently diagnosed with cancer and long-term cancer survivors. We had hoped to isolate individuals based on their cancer diagnosis or survival time, but unfortunately, most of the studies did not provide separate data on these groups. Second, there were only 13 studies included in this meta-analysis, seven studies investigated the risk of dementia among patients with breast cancer, and seven studies explored the risk of breast cancer in individuals with dementia, which may raises concerns about the statistical power and generalizability of the findings. Although, our statistical methods were standardized, the studies have no significant publication bias, and the result was robust suggested by sensitivity analysis, more studies are needed to strengthen the study. Third, there was no subgroup analysis due to the limited number of studies, so factors such as age, follow-up years, survival threshold, breast cancer type, dementia type, or therapy were not examined. Future large-scale population-based studies are needed to better understand the breast cancer-dementia relationship. Forth, variability in treatment plans and follow-up durations among breast cancer could introduce clinical heterogeneity, and differences in the quality of inclusion criteria among studies may lead to methodological heterogeneity. Despite some heterogeneity, our study’s results remain robust, as indicated by sensitivity analyses (Li et al., 2023; Wang et al., 2024).

Conclusions

The result shown that breast cancer was less likely to develop to dementia, and dementia is similarly low associated with risk of breast cancer. Our findings will provide clinicians with essential insights into the prevention and monitoring of neurodegenerative diseases in this patient population. Nonetheless, it is important to acknowledge the limitations of this study.

Supplemental Information

Supplemental Information 1 PRISMA checklist.

Supplemental Information 2 Sensitivity analysis of breast cancer and the risk of dementia.

Supplemental Information 3 Sensitivity analysis of dementia and the risk of breast cancer.

Supplemental Information 4 Publication bias of breast cancer and the risk of dementia.

Supplemental Information 5 Publication bias of dementia and the risk of breast cancer.

Supplemental Information 6 Details of the Literature Search Strategy in PubMed.

Supplemental Information 7 Details of the Literature Search Strategy in Cochrane Library.

Supplemental Information 8 Details of the Literature Search Strategy in Embase.

Supplemental Information 9 The excluded studies.

Supplemental Information 10 Sensitivity analysis for breast cancer and the risk of dementia.

Supplemental Information 11 Sensitivity analysis for dementia and the risk of breast cancer.

Additional Information and Declarations

Competing Interests

The authors have no conflicts of interest to disclose.

Author Contributions

Fuxing Bao conceived and designed the experiments, performed the experiments, analyzed the data, authored or reviewed drafts of the article, and approved the final draft.

Liang Yu conceived and designed the experiments, analyzed the data, authored or reviewed drafts of the article, and approved the final draft.

Xiaolei Zhang analyzed the data, prepared figures and/or tables, and approved the final draft.

Qier Mu performed the experiments, prepared figures and/or tables, and approved the final draft.

Data Availability

The following information was supplied regarding data availability:

This is a systematic review and meta analysis.

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
