# Peer review of "Bidirectional association between breast cancer and dementia: a systematic review and meta-analysis of observational studies"

_PeerJ, doi:10.7717/peerj.18888_

## Round 0.1 · original submission · Major Revisions

· Academic Editor

Major Revisions

Dear Dr. Bao,

Based on the two reviewers' comments, concerns have been raised regarding the statistical power, generalizability of the findings, and the significant differences in the study population used to reach the conclusions. Therefore, we have decided to invite a major revision of your manuscript.

Please provide a point-by-point response to the reviewers' concerns, with the revisions highlighted in the updated manuscript. We also recommend thoroughly proofreading the manuscript, ideally with the assistance of a professional language editor, to address any grammatical or syntax errors.

Thank you for your submission to PeerJ.

Thanks

Abhishek Tyagi
Academic Editor
PeerJ

Reviewer 1 ·

Basic reporting

Overall, the structure of manuscript is fine but English editing is needed throughout (e.g., “here were seven studies mentioned the risk of dementia associated with breast cancer, and shown that dementia is also lower associated with risk of breast cancer”, “The result shown that breast cancer was less likely to develop to dementia, and dementia is also lower associated with risk of breast cancer”).

It is also unclear to me why there are two different abstracts. I see one starting with “Cognitive decline after cancer treatment...” and one starting “Cognitive decline following cancer treatment...”

Experimental design

I think that this is very important subject for a meta-analysis and applaud the authors for tackling what is a field full of methodological complications. However, I have some questions about the methods.

The studies included included here conflate patients with present cancer diagnosis and those who are cancer survivors (for 5 years or more). This is a massive difference in the study population considered and by mixing the two populations together, it is impossible to draw any conclusion about risk of dementia. To be clear, many studies actually conflate these two groups (e.g., Oh et al 2023). However, the difference between those living with cancer/recent survivors and 5-year survivors is vast. It has been well-documented that patients with cancer have lower risk of dementia, but it is unclear what the risk of dementia is among long-term cancer survivors (e.g., follow-up starting 5+ years after cancer diagnosis).

It’s also unclear why the authors include some studies in the background information (e.g., Carriera et al 2021, Wennberg et al 2023) but not in the meta-analysis

Validity of the findings

My concerns about the conclusions relate to the concerns I stated above.

Reviewer 2 ·

Basic reporting

This meta-analysis includes only 13 studies, which raises concerns about the statistical power and generalizability of the findings. While meta-analyses can be conducted with a smaller number of studies, the current number is borderline and may not provide sufficient robustness. The manuscript does not adequately address whether the studies are comparable in terms of study design (only included observational studies). There are more observational studies published recently. More studies are needed to strengthen the study.

The manuscript has several formatting and grammatical errors. For example, "breast cancer is lower associated with risk of dementia" should be revised.

The results are presented ambiguously in places, especially in the discussion of the relationship between breast cancer and dementia. The use of phrases like "lower associated" is unclear, and the confidence intervals cross 1, suggesting no statistically significant association. This should be clarified to avoid misinterpretation.

Experimental design

This meta-analysis includes only 13 studies, which raises concerns about the statistical power and generalizability of the findings. More studies are needed to strengthen the study.

Validity of the findings

The use of phrases like "lower associated" is unclear, and the confidence intervals cross 1, suggesting no statistically significant association. This should be clarified to avoid misinterpretation.

---

## Round 0.2 · Major Revisions

· Academic Editor

Major Revisions

Dear Dr. Bao,

After initial assessment of the revised submission and point-by-point response, I have concluded that the authors have not satisfactorily addressed all reviewer comments. I, therefore, recommend a further major revision of the manuscript.

Warm regards,
Abhishek Tyagi
Academic Editor
PeerJ Life & Environment

Reviewer 2 ·

Basic reporting

no comment

Experimental design

This meta-analysis includes only 13 studies, which raises concerns about the statistical power and generalizability of the findings. More studies are needed to strengthen the study by expanding search strategy and including more relevant and recent studies.

Validity of the findings

no comment

---

## Round 0.3 · Minor Revisions

· Academic Editor

Minor Revisions

Dear Author,

After careful assessment of the revised manuscript, I recommend addressing the reviewer's point by explicitly highlighting the response as a limitation of the study in the section following the discussion.

Best regards,

Abhishek Tyagi
Academic Editor

---

## Round 0.4 · accepted · Accept

· Academic Editor

Accept

Dear Dr. Bao,

Thank you for your revised submission to PeerJ.
I am writing to inform you that your manuscript, Bidirectional association between breast cancer and dementia: a systematic review and meta-analysis of observational studies, has been accepted for publication afetr assessing the revised version.

Congratulations, and thank you for your submission.

Warm regards,

Abhishek Tyagi
Academic Editor
PeerJ Life & Environment